# Dental Anomalies in Ciliopathies: Lessons from Patients with *BBS2*, *BBS7,* and *EVC2* Mutations

**DOI:** 10.3390/genes14010084

**Published:** 2022-12-27

**Authors:** Piranit Kantaputra, Prapai Dejkhamron, Rekwan Sittiwangkul, Kamornwan Katanyuwong, Chumpol Ngamphiw, Nuntigar Sonsuwan, Worrachet Intachai, Sissades Tongsima, Philip L. Beales, Worakanya Buranaphatthana

**Affiliations:** 1Division of Pediatric Dentistry, Department of Orthodontics and Pediatric Dentistry, Faculty of Dentistry, Chiang Mai University, Chiang Mai 50200, Thailand; 2Dentaland Clinic, Chiang Mai 50200, Thailand; 3Center of Excellence in Medical Genetics Research, Faculty of Dentistry, Chiang Mai University, Chiang Mai 50200, Thailand; 4Division of Pediatric Endocrinology, Department of Pediatrics, Faculty of Medicine, Chiang Mai University, Chiang Mai 50200, Thailand; 5Division of Pediatric Cardiology, Department of Pediatrics, Faculty of Medicine, Chiang Mai University, Chiang Mai 50200, Thailand; 6Division of Pediatric Neurology, Department of Pediatrics, Faculty of Medicine, Chiang Mai University, Chiang Mai 50200, Thailand; 7National Biobank of Thailand, National Science and Technology Development Agency, Pathum Thani 12120, Thailand; 8Department of Otolaryngology, Faculty of Medicine, Chiang Mai University, Chiang Mai 50200, Thailand; 9Genetics and Genomic Medicine Program, UCL Great Ormond Street Institute of Child Health, University College London, London WC1N 1EH, UK; 10Department of Oral Biology and Diagnostic Sciences, Faculty of Dentistry, Chiang Mai University, Chiang Mai 50200, Thailand

**Keywords:** ciliopathy, dental anomalies, tooth malformations, Ellis–van Creveld syndrome, Bardet–Biedl syndrome, primary cilia

## Abstract

Objective: To investigate dental anomalies and the molecular etiology of a patient with Ellis–van Creveld syndrome and two patients with Bardet–Biedl syndrome, two examples of ciliopathies. Patients and Methods: Clinical examination, radiographic evaluation, whole exome sequencing, and Sanger direct sequencing were performed. Results: Patient 1 had Ellis–van Creveld syndrome with delayed dental development or tooth agenesis, and multiple frenula, the feature found only in patients with mutations in ciliary genes. A novel homozygous mutation in *EVC2* (c.703G>C; p.Ala235Pro) was identified. Patient 2 had Bardet–Biedl syndrome with a homozygous frameshift mutation (c.389_390delAC; p.Asn130ThrfsTer4) in *BBS7*. Patient 3 had Bardet–Biedl syndrome and carried a heterozygous mutation (c.389_390delAC; p.Asn130ThrfsTer4) in *BBS7* and a homozygous mutation in *BBS2* (c.209G>A; p.Ser70Asn). Her clinical findings included global developmental delay, disproportionate short stature, myopia, retinitis pigmentosa, obesity, pyometra with vaginal atresia, bilateral hydronephrosis with ureteropelvic junction obstruction, bilateral genu valgus, post-axial polydactyly feet, and small and thin fingernails and toenails, tooth agenesis, microdontia, taurodontism, and impaired dentin formation. Conclusions: *EVC2*, *BBS2*, and *BBS7* mutations found in our patients were implicated in malformation syndromes with dental anomalies including tooth agenesis, microdontia, taurodontism, and impaired dentin formation.

## 1. Introduction

The primary cilium is an immotile single organelle found on the apical surface of the majority of cells in the human body [1,2]. During the development of vertebrates, primary cilia play a key role in sensory perception and coordinating various signaling pathways including Hh, Wnt, and PDGFR signaling [3,4,5]. Defects in cilia biogenesis or intraflagellar transport (IFT) impair diverse physiological functions including motility-related and sensory-related ciliary functions and result in a number of human diseases termed “ciliopathies” [1,6,7]. Ciliopathies, the defects of either structure or function of cilia, are a large group of genetic disorders including Ellis–van Creveld syndrome (EVC; MIM 225500), Bardet–Biedl syndromes (BBS), Weyers acrofacial dysostosis (WAD; MIM 193530), cranioectodermal dysplasia (CED1; MIM 218330), McKusick–Kaufman syndrome (MKKS; MIM 236700), and orofacialdigital syndrome (OFD1; MIM 311200) [2,3]. These ciliopathies share a number of clinical phenotypes including dental anomalies, among other things. Ellis–van Creveld syndrome is an autosomal recessive disorder, caused by mutations in *EVC* or *EVC2*. Characteristic features of EVC consist of rhizomelic dysplasia, short ribs, postaxial polydactyly, dental anomalies, multiple frenula, and nail dysplasia. Bardet–Biedl syndrome, an autosomal recessive multisystem disorder, is a genetically heterogeneous developmental disorder characterized by rod-cone-dystrophy, obesity, genital anomalies, kidney abnormalities, learning difficulties, polydactyly, and dental anomalies [8,9,10]. During tooth development, primary cilia, which are located in the dental epithelium and mesenchyme of the developing tooth germs, sense and transduce signals during epithelial–mesenchymal interactions. The length of the cilia depends on the function of the cells. Cilia are longer in the cells of signaling centers such as the enamel knots or cells that undergo intense cytodifferentiation and the production of dentin or enamel [11]. Ciliary dysfunction has been shown to cause supernumerary teeth, tooth agenesis, microdontia, taurodontism, and enamel and dentin hypoplasia [8,9,11].

Here, we report three Thai patients, one with EVC and two patients with BBS. Patient 1 with EVC carried a novel homozygous mutation in *EVC2* (MIM 607261). Patient 2 with BBS carried a homozygous mutation in *BBS7.* Patient 3, also with BBS, carried a homozygous mutation in *BBS2* as well as a heterozygous mutation in *BBS7*. Ciliopathy-associated signaling pathways leading to dental anomalies in patients with BBS and EVC are discussed.

## 2. Patients and Methods

The study was conducted in accordance with the Declaration of Helsinki and the national guidelines. Informed consent was obtained from the parents in accordance with the regulations of the Human Experimentation Committee of the Faculty of Dentistry, Chiang Mai University (Certificate of Ethical Clearance number 25/2020).

### 2.1. Subjects


**Patient 1**


A 2-year-old boy, presenting with heart failure and failure to thrive, was referred from a provincial hospital to Chiang Mai University Hospital for cardiac surgery (Figure 1A). He was the fourth child of a non-consanguineous Thai couple. Parents were healthy with normal dentitions. Their second child was female, who died a few weeks after birth (Figure 1B). She had a common atrium, post-axial polydactyly of both hands, and hypotrichosis. The first child (female) and the third child (male) were normal. Physical examination revealed mild cyanosis (oxygen saturation 82%), a systolic ejection murmur grade II/VI at the left parasternal border and apex, a long narrow chest, rhizomelic shortening of the limbs, post-axial polydactyly of both hands, and hypoplastic nails (Figure 1E,F). Intraoral examination showed no primary teeth (possible tooth agenesis or delayed tooth eruption) and multiple frenula with high attachment (Figure 1C,D). The patient was too young for radiographic examination.

Growth and development assessment showed delayed gross motor development. Chest radiography showed cardiomegaly with increased pulmonary blood flow. Echocardiography and cardiac computed tomography demonstrated a common atrium, cleft mitral valve, severe mitral regurgitation, small patent ductus arteriosus (PDA), bilateral superior vena cava (SVC), and left SVC draining into the left atrium and a patent ductus arteriosus (Figure 1G,H). He underwent atrial septal defect closure, patent ductus arteriosus ligation, mitral valve repair, and connection of the left SVC to right atrium. He experienced a chylothorax during the post-operative period and was later discharged. In his last out-patient appointment, his clinical heart failure improved with normal weight gain but he still had developmental delay.


**Patient 2**


A 1.5-month-old boy was admitted to Chiang Mai University Hospital with the chief complaints of dyspnea and central cyanosis for 2 weeks (Figure 2A). He was diagnosed with cyanotic heart disease and bacterial septicemia and treated with intravenous antibiotics and placed on respiratory support. He was born at home and his parents originated from a Northern Thai hill tribe and lived in the same village. His parents and elder brother were healthy with normal dentitions (Figure 2B). His weight was 3.3 kg (<3 centile). He had central cyanosis with multiple anomalies including a macular hemangioma on the forehead, bulbous nose, cup-shaped ears, micrognathia, polydactyly of all extremities, and syndactyly of the left hand (Figure 2A–E). He had no murmur and a single S_2_ heart sound at the right side of his chest. Echocardiography and cardiac computed tomography revealed complex cyanotic heart disease and a single ventricle physiology with ductal-dependent pulmonary circulation (dextrocardia, tricuspid atresia, atrial septal defect, ventricular septal defect, pulmonary atresia, non-confluent pulmonary artery, and patent ductus arteriosus with collateral vessels) (Figure 2F). He underwent a central systemic to pulmonary shunt and pulmonary artery plasty. During the post-operative period, he had a prolonged ICU admission due to acute renal failure and systemic candidiasis. He was discharged to a provincial hospital 5 weeks after surgery. Unfortunately, he died 1 week after being discharged. The patient was too young for dental evaluation.


**Patient 3**


A 12-year-old Thai girl was born to non-consanguineous parents. Both parents were healthy with unremarkable dentitions. Her father passed away a few years ago. She presented with short stature, night-blindness, and bilateral knee pain (Figure 3A,B). She was born following an uneventful pregnancy at full term with a birthweight of 2.95 kg. Her neonatal and infancy period was complicated by pyometra with vaginal atresia, which required colpostomy with percutaneous drainage and vaginoplasty. She also experienced recurrent urinary tract infections. Investigations revealed bilateral hydronephrosis with ureteropelvic junction obstruction.

Her developmental milestones were globally delayed. Physical examination revealed a weight of 38.3 kg (P50), height 121 cm (<P3), and upper/lower segment ratio 1.2:1 (normal 1:1), indicating disproportionate short stature, body mass index (BMI) of 26.2 kg/m^2^, obesity, bilateral genu valgus, and post-axial polydactyly of the feet. Fingernails and toes nails were small (Figure 3C,D). Eye examination revealed myopia and retinal dystrophy, which was consistent with early retinitis pigmentosa. Oral manifestations consisted of agenesis of the mandibular left permanent first molar and microdontia of the mandibular premolars (Figure 3E). Panoramic radiograph showed taurodontism of all permanent molars, absence of the mandibular left permanent first molar, and generalized large root canal spaces, indicating impaired dentin formation (Figure 3F).

### 2.2. Whole Exome Sequencing and Mutation Analysis

Genomic DNA was extracted from blood according to the standard procedure. The DNA samples of the patient and unaffected parents or siblings were exome-sequenced with the SureSelect V6 + UTR-Post Target Capture Kit. Genomics Analysis Toolkit (GATK) germline mutation workflow version 3.8.1 [12] was utilized to identify variants. The sequencing reads were aligned to hg19 using BWA-MEM version 0.7.17 [13] to generate BAM files. These BAM files were processed by GATK HaplotypeCaller to identify SNVs and small indels, resulting in individual GVCF files. These GVCF files were consolidated into a single joint-genotyped VCF file format, listing all genotypes in separate columns. Ensembl Variant Effect Predictor Tool (version 95) [14] was used to predict the pathogenic effects of those variants.

## 3. Results

### Whole Exome Sequencing and Mutation Analysis


**Patient 1**


Whole exome sequencing showed a homozygous missense mutation in *EVC2* (chr4:g.5667304C>G; NM_001166136.1:c.703G>C; NP_001159608.1:p.Ala235Pro) in the patient. The unaffected parents and the unaffected elder brother were heterozygous for the mutation (Figure 4).


**Patient 2**


Whole exome sequencing showed a homozygous two-base deletion mutation (chr4:g.122780285G>T; NM_018190.3:c.389_390del; NP_789794.1:p.Asn130ThrfsTer4; rs863224530) in *BBS7*. This base deletion is predicted to result in an amino acid residue change from Asn to Thr at amino acid position 130 and subsequent truncation of the protein four amino acids later (p.Asn130ThrfsTer4) (Figure 4).


**Patient 3**


Whole exome sequencing showed a homozygous mutation in *BBS2* (chr16:g. 56548501C>T; NM_031885.5; c.209G>A; NP_114091.4; p.Ser70Asn; rs4784677) accompanied by a heterozygous base deletion NM_018190.3:c.389_390del (p.Asn130rsTer4) in *BBS7* (Figure 4). Sanger direct sequencing confirmed the variants in patients 1–3 (Figure 4).

## 4. Discussion

We report three patients, one with EVC and two with BBS. The EVC patient (patient 1) was a boy who had a novel homozygous mutation in *EVC2* (c.703G>C; p.Ala235Pro). This mutation was predicted to be possibly damaging, polymorphism, and tolerated by PolyPhen-2, MutationTaster, and SIFT. His oral features were tooth agenesis or delayed dental development, and multiple frenula. His sister, who died a few weeks after birth, also had EVC with a common atrium and postaxial polydactyly. Patient 2, who was affected with BBS, carried a homozygous mutation (c.389_390delAC; p.Asn130ThrfsTer4) in *BBS7*. This mutation was predicted to be disease causing by MutationTaster. He was too young to have a dental evaluation. Patient 3 was affected with BBS with tooth agenesis, microdontia, taurodontism, and impaired dentin formation. She was heterozygous for the *BBS7* variant (c.389_390delAC; p.Asn130ThrfsTer4), the variant found in patient 2. Interestingly, patients 2 and 3 shared the rare p.Asn130ThrfsTer4 variant. Although parents of both parents were not aware that they were related, it is important to note that these parents and their families were Hmong living in Thailand. This suggests a possible founder effect. In addition to the heterozygous *BBS7* frameshift mutation, which has been previously described in BBS patients, patient 3 also carried a homozygous variant in *BBS2* (c.209G>A; p.Ser70Asn; rs4784677). According to gnomAD, this *BBS2* variant is common with an allele frequency of 0.9944. It is considered benign by Varsome and ACMG Classification. Interestingly, this variant has been reported in previous cases with a digenic triallelic inheritance of BBS. BBS is a complex genetic disease, which in some patients require two mutations in one *BBS* gene, accompanied by a third *BBS* gene variant, which is considered as a contributing factor to manifest the BBS phenotype [15]. However, this is the first time that that two purportedly benign variants (*BBS2*) have been seen in combination with a heterozygous pathogenic mutation (*BBS7*). We therefore hypothesize that the phenotype in patients 2 and 3 are likely to be the consequences of stoichiometric disruption of the BBSome complex in the cilium as both BBS2 and BBS7 proteins are important components [16,17].

### 4.1. BBS2 and BBS7 Mutations and Impaired BBSome Complex

BBS2 and BBS7 are two of the eight highly conserved BBS proteins that form the BBSome, the heterooctameric protein complex required for ciliary membrane biogenesis. BBS7 is a unique subunit that stabilizes BBS2 in the BBSome and has a direct physical interaction with the BBS chaperonin complex (made up of BBS6, BBS10, and BBS12). The presence of a mutant subunit in a multi-subunit complex like the BBSome results in the malfunction of the entire complex [18,19]. BBS7 has an important role in the localization of the ciliary membrane proteins and specific ciliary membrane trafficking. Mice lacking in *Bbs7* have primary cilia and IFT complexes [17], but there is an abnormal accumulation of the dopamine D1 receptor to the ciliary membrane as a result of abnormal IFT. Therefore in the case of patient 3, although the variant in *BBS2* (p.Ser70Asn) is predicted to generally be benign, it is hypothesized that the accompanying *BBS7* frameshift mutation (albeit heterozygous) is enough to upset the stoichiometric balance of the BBSome by “pushing” the *BBS2* p.Ser70Asn beyond a threshold of pathogenicity. Absence of *BBS7* also results in the accumulation of Smo within the cilia [20,21], which results in aberrant Hh signaling and subsequent Hh signaling phenotypes including dental anomalies (Figure 5).

### 4.2. Oral Manifestations in EVC and BBS

Both EVC and BBS share a number of clinical findings including dental anomalies, the malformations related to cilia-associated impaired Hh and Wnt signaling. Patient 3, who was heterozygous for a *BBS7* variant and homozygous for a *BBS2* variant, had tooth agenesis, microdontia, taurodontism, and large dental pulp spaces, indicating impaired dentin formation. Oral manifestations of patient 1 consisted of tooth agenesis or delayed tooth eruption, and multiple frenula. Regarding multiple frenula found in patient 1, who had EVC, to the best of our knowledge, multiple frenula are found only in patients affected with ciliopathies including *EVC* or *EVC2*-associated EVC, *OFD1*-associated orofaciodigital syndrome, GLI3-associated Pallister–Hall syndrome, and CEP120-associated short rib thoracic dysplasia 13 with or without polydactyly, CEP120-associated Joubert syndrome, and WDR35-associated short rib thoracic dysplasia 7 with or without polydactyly [22]. The current knowledge on the embryology of the frenulum is not sufficient to explain why only mutations in ciliary genes cause multiple frenula. Interestingly, the syndromes that have multiple frenula as a feature also have natal teeth as a feature.

### 4.3. EVC, BBS, and Abnormal Hh and Wnt Signaling

Hh and Wnt signaling are crucial for tooth development. Mutations in both signaling pathways are known to cause various kinds of human malformations including dental anomalies. Cilia promotes Hh signaling, but restrains canonical Wnt signaling, indicating their functional antagonism is mediated by cilia [23,24,25]. Loss of BBS protein function results in abnormalities in the suppression of non-canonical Wnt signaling, with an increase in targets of canonical Wnt signaling [26]. Therefore, mutations in *BBS* genes would result in malformations similar to those with the upregulation of canonical Wnt signaling [27] (Figure 5).

Regarding tooth development, *EVC2*, *BBS2*, and *BBS7* mutations found in our patients are predicted to result in abnormal BBSome function, disrupted intraflagellar transport, abnormal cilia biogenesis and function, impaired Hh and Wnt signaling, and subsequent dental anomalies including tooth agenesis, microdontia, taurodontism, and impaired dentin formation, among other phenotypes (Figure 5). In vertebrates, Hh signaling is initiated at the primary cilia by the ligand-triggered accumulation of Hh signaling effector protein Smoothened (Smo) in the ciliary membrane, which is absolutely required for Hh signaling in all tissues [28]. EVC and EVC2, positive tissue-specific regulators of Hh signaling pathways, form the EVC–EVC2 complex and acts on Smo in Hh signal transduction [29]. The accumulation of Smo in cilia in response to Hh signaling results in a physical association of Smo and EVC2 in the EVC–EVC2 complex in the EVC zone [28]. The EVC–EVC2 complex localizes in the basal body of the primary cilia by anchoring itself with ciliary protein complex EFCAB7–IQCE in a signaling microdomain. EFCAB7 functions as an adaptor protein linking IQCE to the EVC–EVC2 complex at the C-terminal disordered region in EVC2. The role of EVC2 in Hh signaling is to translate Smo activation to the inhibition of SuFu and PKA (Figure 5). Therefore, localization of the EFCAB7–IQCE protein complex in the basal body of cilia is a requirement for Hh signaling [29,30] and mutations in *EVC2* are predicted to cause impaired linking of the EFCAB7–IQCE complex to the EVC–EVC2 complex, which results in the dysregulation of Hh signaling because the Smo–EVC2 signaling complex at the EVC zone is required for Hh signal transduction (Figure 5).

EVC and EVC2 are required for Hh signaling, therefore, *Evc*-like genes are absent in organisms that do not have Hh signaling (worms) or do not possess cilia for Hh signaling (flies). Patients with *EVC2* mutations have clinical findings of abnormal Hh signaling including dental anomalies because in patients with *EVC2* mutations, EVC2 proteins are not functional because they fail to localize in the cilia, and function as dominant inhibitors of Hh signaling [28]. Mice lacking *Evc* or *Evc2* have defective biomineralization [31]. In humans, patients with mutations in *EVC* and *EVC2* have impaired Hh signaling phenotypes including congenital heart defects, limb anomalies, and dental anomalies [28]. *Evc-/-* mice have microdontia and the fusion of roots [32,33,34]. The phenotypes of the patients with *EVC2* mutations are associated with the failure of signaling complex EVC2–Smo to assemble in the EVC zone [28] (Figure 5).

### 4.4. Ciliopathy, WNT Signaling, and Dental Anomalies

In addition to Bardet–Biedl syndrome and Ellis–van Creveld syndrome, dental anomalies have been reported in other ciliopathy-associated syndromes including Weyers acrofacial dysostosis (MIM 193530), Joubert syndrome (MIM 213300), primary cilia dyskinesis (MIM 244400), cranioectodermal dysplasias (MIM 218330, 614099, 613610, 614378), and orofaciodigital syndrome I (MIM 311022) ([35] Shen et al., 2011; [36] Pawlaczyk-Kamieńska et al., 2020).

When the Wnt ligand binds to its receptor complex, it inhibits the function of the β-catenin destruction complex, allowing for the accumulation of β-catenin in the cytoplasm and its nuclear translocation as well as the transcription activation via T-cell factor/lymphoid enhancer-binding factor transcription factors. Loss of Hh signaling generally leads to the overactivation of Wnt signaling, and impaired ciliary function in ciliary mutants [24]. Non-canonical and canonical Wnt signaling are associated with the cilia and basal body. BBS proteins regulate proteasome function and transport Inversin from the basal body to the cytoplasm, where it reduces the cytoplasmic levels of disheveled protein (DVL) via phosphorylation. Mutations in *BBS* disrupt retrograde IFT and disrupt the transport of Inversin to the cytoplasm, preventing its interaction with DVL [27,37] (Figure 5).

Deletion of IFT140, a subunit of IFT complex A, which is important for retrograde transportation of cilia, leads to poor odontogenic differentiation, abnormal primary cilia, decreased Shh signaling, and poor dentin formation [38]. The thinning of dentin in patient 3, who had mutations in *BBS7* and *BBS2*, suggests that its pathogenetic mechanism is similar to the deletion of *IFT140* or the abnormal retrograde transportation of cilia [39]. Primary cilia regulate Shh activity in the control of tooth number. Mice mutant of IFT88/Polaris show an upregulation of Shh activity, and subsequent ectopic tooth formation [40].

**Figure 5 genes-14-00084-f005:**
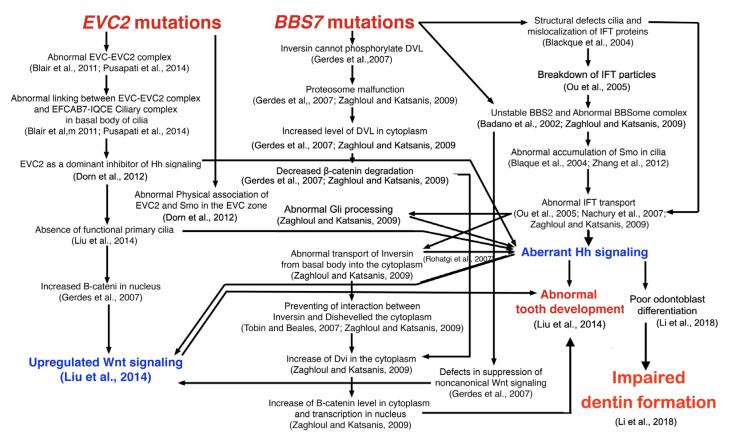
Flowchart shows that *EVC2*, *BBS2*, and *BBS7* mutations resulted in an abnormal BBSome complex, disrupted intraflagellar transport, showed abnormal cilia biogenesis and function, impaired Hh and Wnt signaling, and subsequent dental anomalies including hypodontia, microdontia, taurodontism, and impaired dentin formation [16,18,20,21,25,26,27,28,29,30,38,41,42,43].

## 5. Conclusions

The *EVC2*, *BBS2*, and *BBS7* mutations found in our patients are predicted to result in an abnormal BBSome complex, disrupted intraflagellar transport, abnormal cilia biogenesis and function, impaired Hh and Wnt signaling, and subsequent dental anomalies including tooth agenesis, microdontia, taurodontism, and impaired dentin formation. The phenotypes of the patients with *EVC2* mutations are associated with the failure of signaling complex EVC2–Smo to assemble in the EVC zone. This is the first time that two benign *BBS2* variants have been seen in combination with a heterozygous pathogenic *BBS7* mutation. It is hypothesized that the BBS phenotypes in our patients are the likely consequences of the stoichiometric disruption of the BBSome complex in the cilium as both BBS2 and BBS7 proteins are important components.

## Figures and Tables

**Figure 1 genes-14-00084-f001:**
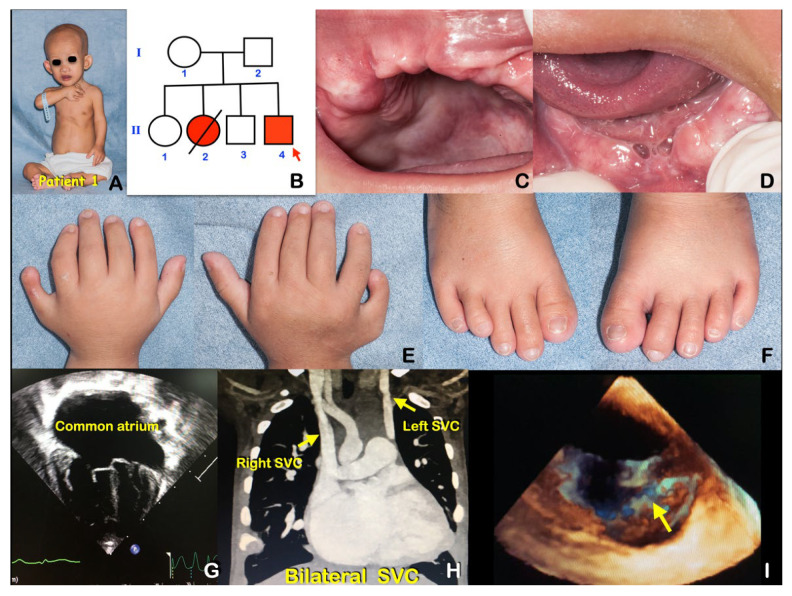
Patient 1 at age 2 years. Ellis–van Creveld syndrome caused by a novel homozygous *EVC2* mutation. (**A**) Note hypotrichosis and narrow chest. (**B**) Pedigree shows that case II-2 was affected with Ellis–van Creveld syndrome and died. (**C**,**D**) Multiple frenula and serrated alveolar ridges. (**E**) Post-axial polydactyly of the hands. Note small fingernails. (**F**) Toenails were small. (**G**) Echocardiogram showed a common atrium. (**H**) Echocardiogram showed bilateral superior vena cava. (**I**) Three-dimensioned echocardiogram showed cleft mitral valve.

**Figure 2 genes-14-00084-f002:**
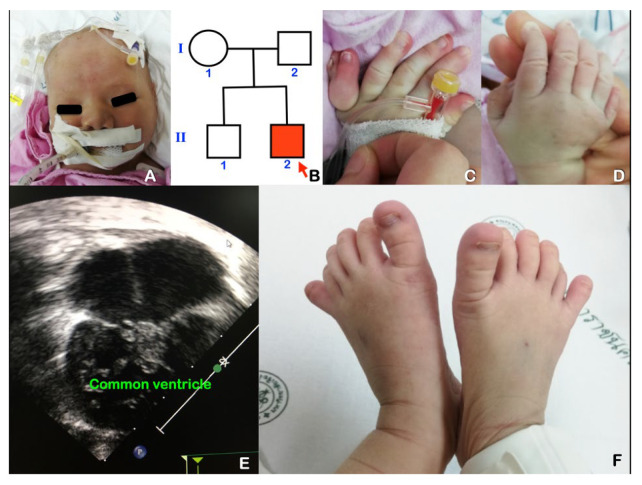
Patient 2 at age 1.5 months had Bardet–Biedl syndrome caused by a homozygous mutation in *BBS7*. (**A**) Bulbous nose and micrognathia. (**B**) Pedigree. (**C**,**D**) Post-axial polydactyly of the hands. (**E**) Echocardiogram showed a common ventricle. (**F**) Post-axial polydactyly of the feet.

**Figure 3 genes-14-00084-f003:**
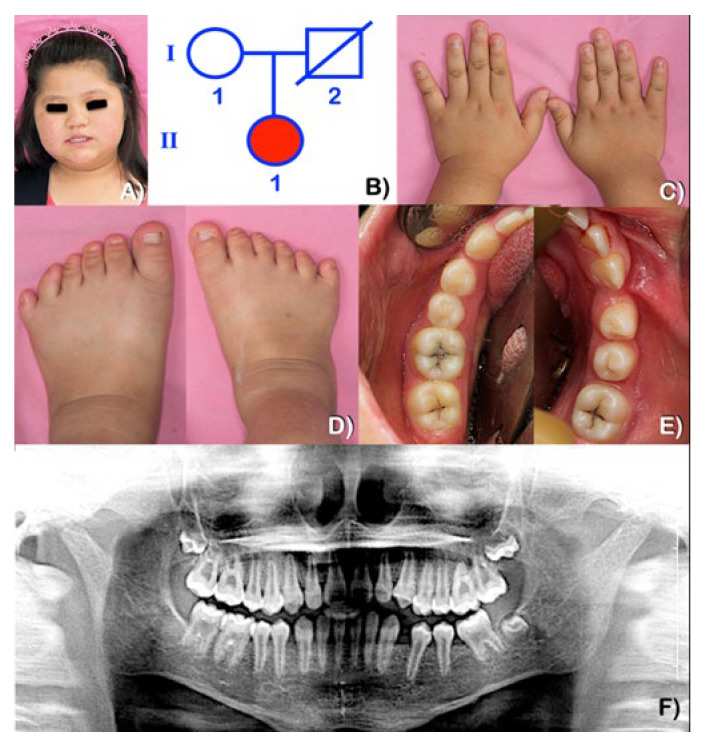
Patient 3 at age 12 years has Bardet–Biedl syndrome caused by digenic triallelic mutation in *BBS2* and *BBS7*. (**A**) Obesity. (**B**) Pedigree. (**C**) Small fingernails. (**D**) Post-axial polydactyly of the feet. Note small toenails. (**E**) Microdontia of the mandibular premolars. Note the dental caries on the permanent molars. (**F**) Panoramic radiograph showed the absence of the mandibular left first permanent molar, microdontia of the mandibular left third permanent molar, taurodontism of all permanent molars, and large root canal spaces, indicating impaired dentin formation.

**Figure 4 genes-14-00084-f004:**
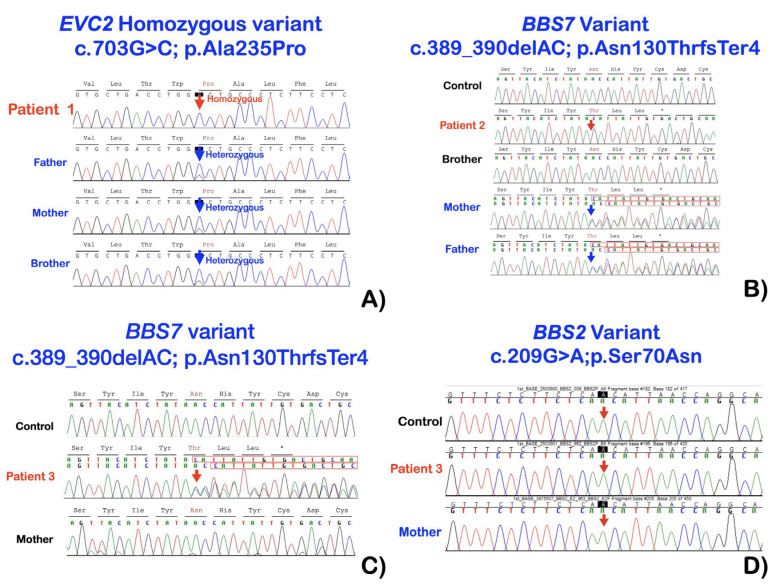
(**A**) Patient 1. Electropherograms showed a homozygous missense mutation in *EVC2* (NM_001166136.1:c.703G>C; NP_001159608.1:p.Ala235Pro). The unaffected parents and the unaffected elder brother were heterozygous for the mutation. (**B**) Patient 2. Electropherograms showed a homozygous two-base deletion mutation NM_018190.3:c.389_390del (p.Asn130ThrfsTer4) in *BBS7*. This base deletion is predicted to result in an amino acid residue change from Asn to Thr at amino acid position 130 and subsequent truncation of the protein four amino acids later (p.Asn130ThrfsTer4). (**C**) Patient 3. Electropherograms showed a heterozygous two-base deletion mutation NM_018190.3:c.389_390del (NP_789794.1:p.Asn130ThrfsTer4) in *BBS7*. This base deletion is predicted to result in an amino acid residue change from Asn to Thr at amino acid position 130 and subsequent truncation of the protein four amino acids later (p.Asn130ThrfsTer4). (**D**) Patient 3. Electropherograms showed a homozygous missense mutation in *BBS2* (c.209G>A; p.Ser70Asn; rs4784677).

## Data Availability

No new data were created.

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
