# Peer review of "Dental Anomalies in Ciliopathies: Lessons from Patients with BBS2, BBS7, and EVC2 Mutations"

_genes, 2022, doi:10.3390/genes14010084_

Round 1

Reviewer 1 Report

The authors describe dental anomalies in human associated with ciliopathies, which is interesting and important. The manuscript is well prepared and I enjoy reading it. Although I am recommending a major revision due to the number of issues that I hope the authors could address, these issues should not be hard to address. I am looking forward to the authors’ responses.

1. Title. Why not include BBS2?

2. In the conclusion session of the abstract and the conclusion session at the end of the manuscript, the authors include their predicted mechanisms. I think the conclusion session should be limited to brief statements that are supported by solid direct evidence. Since there is no direct evidences to support these prediction, I don’t think they belong to the conclusion session. 

3. Patient 1. At this stage, there is no erupted primary teeth or radiographic examination, we don’t know whether it is tooth agenesis or delayed tooth development/eruption. Therefore, calling it “tooth agenesis” could be wrong. Please revise lines 95 and 212 accordingly.

4. Patient 1. Please talk about any findings, especially the oral/dental findings, of patient 1’s parents, even if they are relatively normal. 

5. Patient 2. Please talk about any findings, especially the oral/dental findings, of patient 1’s parents and elder brother, even if they are relatively normal.

6. Patient 3. Pedigree is missing. 

7. Patient 3. Please talk about any findings, especially the oral/dental findings, of patient 1’s parents, even if they are relatively normal.

8. Patients 2&3. Since they share the same BBS7 mutations, are they relatives? Could the authors mention their geographic distance?

9. Line 173 talks about using Ensembl tool to predict pathogenic effects of the mutations. This is important, especially the authors have mentioned the potentially benign BBS2 mutation. The authors should show the predicting pathogenic effects of all the three mutations. 

10. Figure 4. The resolution of the images is too low. The authors need to improve the resolution so that the readers can read all the chromatograms/electropherograms and the deduced amino acid sequences. 

11. For all the mutations, please specify the NM and NP numbers, since there are more than one transcript variants and isoforms. 

12. For the BBS7 mutation in patients 2&3, it is a 2 base deletion, not a single base deletion. Please correct it throughout the text.

13. For patient 3, the authors need to show the corresponding BBS2 chromatogram/electropherograms of the mother and a control. This is important since this mutation is predicted to be benign. 

14. Lines 224 & 329. The BBS2 mutations, though homozygous, should be treated as one variant but biallelic. 

15. In the discussion, the authors should review tooth-associated ciliopathies in human. This can include some specific mutations in some genes cause specific dental/oral problems. For example, EVC2 compound heterozygous mutations cause hypodontia and more in humans (DOI 10.1002/ajmg.a.34125). 

Author Response

Response to comments of Reviewer 1

The authors describe dental anomalies in human associated with ciliopathies, which is interesting and important. The manuscript is well prepared and I enjoy reading it. Although I am recommending a major revision due to the number of issues that I hope the authors could address, these issues should not be hard to address. I am looking forward to the authors’ responses.

 RESPONSE

We are grateful to your kind words towards our paper.  We thank you for the valuable comments and suggestions

  1. Title. Why not include BBS2?

 RESPONSE

Dental anomalies in ciliopathies: Lessons from patients with BBS2, BBS7 and EVC2 mutations. Thank you for this comment. The BBS2 variant is discussed in the discussion as followed.

In addition to the heterozygous BBS7 frameshift mutation which has been previously described in BBS patients, patient 3 also carried a homozygous variant in BBS2 (c.209G>A; p.Ser70Asn; rs4784677). According to gnomAD, this BBS2 variant is common with allele frequency of 0.9944. It is considered benign by Varsome and ACMG Classification. Interestingly this variant has been reported in previous cases with digenic triallelic inheritance of BBS. BBS is a complex genetic disease which in some patients require two mutations in one BBS gene accompanied by a third BBS gene variant which is considered a contributing factor to manifest the BBS phenotype [15].

  1. In the conclusion session of the abstract and the conclusion session at the end of the manuscript, the authors include their predicted mechanisms. I think the conclusion session should be limited to brief statements that are supported by solid direct evidence. Since there is no direct evidences to support these prediction, I don’t think they belong to the conclusion session. 

 RESPONSE

Thank you for this comment. It has been corrected as. Conclusion: EVC2, BBS2, and BBS7 mutations found in our patients were implicated in malformation syndromes with dental anomalies including hypodontia, microdontia, taurodontism, and impaired dentin formation

  1. Patient 1. At this stage, there is no erupted primary teeth or radiographic examination, we don’t know whether it is tooth agenesis or delayed tooth development/eruption. Therefore, calling it “tooth agenesis” could be wrong. Please revise lines 95 and 212 accordingly.

 RESPONSE

Thank you for this comment. We also corrected Line 256 as well.

They have been corrected as

Line 95: Intraoral examination showed no primary teeth (possible tooth agenesis or delayed tooth eruption)

Line 217: His oral features were tooth agenesis or delayed dental development, and multiple frenula.

Line 264: Oral manifestations of patient 1 consist of tooth agenesis or delayed tooth eruption and multiple frenula.

  1. Patient 1. Please talk about any findings, especially the oral/dental findings, of patient 1’s parents, even if they are relatively normal. 

 RESPONSE

It was added as .. Parents were healthy with normal dentitions.

  1. Patient 2. Please talk about any findings, especially the oral/dental findings, of patient 1’s parents and elder brother, even if they are relatively normal.

 RESPONSE

It was added as …..His parents and elder brother were healthy with normal dentitions (Fig. 2B).

  1. Patient 3. Pedigree is missing. 

 RESPONSE

Pedigree is included as Figure 3B. Thank you.

  1. Patient 3. Please talk about any findings, especially the oral/dental findings, of patient 1’s parents, even if they are relatively normal.

 RESPONSE

Thank you for this comment. It was corrected as…A 12-year-old Thai girl was born to nonconsanguineous parents. Both parents were healthy with unremarkable dentitions. Her father passed away a few years ago. She presented with short stature, night-blindness and bilateral knee pain (Fig. 1A).

  1. Patients 2&3. Since they share the same BBS7 mutations, are they relatives? Could the authors mention their geographic distance?

 RESPONSE

Thank you for this comment. The following sentences have been added in the discussion (Lines 227-228).Interestingly, patients 2 and 3 shared the rare p.Asn130ThrfsTer4 variant. Although parents of both parents were not aware that they were related, it is important to note that these parents and their families were Hmong living in Thailand. This suggests a possible founder effect.

  1. Line 173 talks about using Ensembl tool to predict pathogenic effects of the mutations. This is important, especially the authors have mentioned the potentially benign BBS2 mutation. The authors should show the predicting pathogenic effects of all the three mutations. 

 RESPONSE

Thank you for this comment. We added the predictions of mutations in the discussion.

For the EVC2 mutation: This mutation is predicted to be possibly damaging, polymorphism, and tolerated by PolyPhen-2, MutationTaster, and SIFT.

Regarding BBS7 mutation: This mutation is predicted to be disease causing by MutationTaster.

Regarding BBS2 mutation: It is discussed in the discussion as followed.

In addition to the heterozygous BBS7 frameshift mutation which has been previously described in BBS patients, patient 3 also carried a homozygous variant in BBS2 (c.209G>A; p.Ser70Asn; rs4784677). According to gnomAD, this BBS2 variant is common with allele frequency of 0.9944. It is considered benign by Varsome and ACMG Classification. Interestingly this variant has been reported in previous cases with digenic triallelic inheritance of BBS. BBS is a complex genetic disease which in some patients require two mutations in one BBS gene accompanied by a third BBS gene variant which is considered a contributing factor to manifest the BBS phenotype [15].

  1. Figure 4. The resolution of the images is too low. The authors need to improve the resolution so that the readers can read all the chromatograms/electropherograms and the deduced amino acid sequences. 

 RESPONSE

Thank you. We corrected that.

  1. For all the mutations, please specify the NM and NP numbers, since there are more than one transcript variants and isoforms. 

 RESPONSE

They have been included into the paper.

EVC2 variant: (chr4:g.5667304C>G; NM_001166136.1:c.703G>C; NP_001159608.1:p.Ala235Pro)

BBS7 variant: chr4:g.122780285G>T; NM_018190.3:c.389_390del; NP_789794.1:p.Asn130ThrfsTer4; rs863224530

BBS2 variant: chr16:g. 56548501C>T; NM_031885.5; c.209G>A; NP_114091.4; p.Ser70Asn; rs4784677

  1. For the BBS7 mutation in patients 2&3, it is a 2 base deletion, not a single base deletion. Please correct it throughout the text.

 RESPONSE

Thank you for this comment. It has been corrected throughout the paper.

  1. For patient 3, the authors need to show the corresponding BBS2 chromatogram/electropherograms of the mother and a control. This is important since this mutation is predicted to be benign. 

 RESPONSE

The BBS2 variants of the mother and control are shown in Figure 4. Thank you for this comment.

  1. Lines 224 & 329. The BBS2 mutations, though homozygous, should be treated as one variant but biallelic. 

 RESPONSE

Thank you for this comment. Patient 3 were homozygous for the single BBS2 variant.

Line 238: However, this is the first time that that two purportedly benign variants (BBS2) have been seen in combination with a heterozygous pathogenic mutation (BBS7). We therefore hypothesize that the phenotype in patients 2 and 3 are likely the consequences of stoichiometric disruption of the BBSome complex in the cilium as both BBS2 and BBS7 proteins are important components [16, 17].

Line 348: This is the first time that two benign BBS2 variants have been seen in combination with a heterozygous pathogenic BBS7 mutation.

  1. In the discussion, the authors should review tooth-associated ciliopathies in human. This can include some specific mutations in some genes cause specific dental/oral problems. For example, EVC2 compound heterozygous mutations cause hypodontia and more in humans (DOI 10.1002/ajmg.a.34125). 

 RESPONSE

Thank you for this comment. We added this paragraph in the discussion.

4.4. Ciliopathy, WNT signaling, and dental anomalies

In addition to Bardet-Biedl syndromes and Ellis-van Creveld syndrome, dental anomalies have been reported in other ciliopathy-associated syndromes including Weyers acrofacial dysostosis (MIM 193530), Joubert syndrome (MIM 213300), primary cilia dyskinesis (MIM 244400), cranioectodermal dysplasias (MIM 218330, 614099, 613610, 614378), and orofaciodigital syndrome I (MIM 311022) (Shen et al., 2011; Paw-laczyk-Kamieńska et al., 2020).

And these two references are added.

Shen W, Han D, Zhang J, Zhao H, Feng H. Two novel heterozygous mutations of EVC2 cause a mild phenotype of Ellis-van Creveld syndrome in a Chinese family. Am J Med Genet A. 2011 Sep;155A(9):2131-6. doi: 10.1002/ajmg.a.34125.

Pawlaczyk-Kamieńska T, Winiarska H, Kulczyk T, Cofta S. Dental Anomalies in Rare, Genetic Ciliopathic Disorder-A Case Report and Review of Literature. Int J Environ Res Public Health. 2020 Jun 17;17(12):4337. doi: 10.3390/ijerph17124337.

  1. Shen W, Han D, Zhang J, Zhao H, Feng H. Two novel heterozygous mutations of EVC2 cause a mild phenotype of Ellis-van Creveld syndrome in a Chinese family. Am J Med Genet A. 2011 Sep;155A(9):2131-6. doi: 10.1002/ajmg.a.34125.
  2. Pawlaczyk-Kamieńska T, Winiarska H, Kulczyk T, Cofta S. Dental Anomalies in Rare, Genetic Ciliopathic Disorder-A Case Report and Review of Literature. Int J Environ Res Public Health. 2020 Jun 17;17(12):4337. doi: 10.3390/ijerph17124337.

  1. Tobin JL, Beales PL. Bardet-Biedl syndrome: beyond the cilium. Pediatr Nephrol. 2007 Jul;22(7):926-36. doi: 10.1007/s00467-007-0435-0.
  2. Ou G, Blacque OE, Snow JJ, Leroux MR, Scholey JM. Functional coordination of intraflagellar transport motors. Nature. 2005 Jul 28;436(7050):583-7. doi: 10.1038/nature03818.
  3. Rohatgi R, Milenkovic L, Scott MP. Patched1 regulates hedgehog signaling at the primary cilium. Science. 2007 Jul 20;317(5836):372-6. doi: 10.1126/science.1139740.

Reviewer 2 Report

Authors reported clinical case study of three patients containing EVC2, BBS2 and BBS7 mutations, and predicted the mechanism of disease occurrence. The results seem to be interesting from the clinical point of view. However, the proposed mechanism is not conclusive from the viewpoint of experimental science.  Authors should brush up the paper following the comments shown below. 

Major comments:

1. Abstract

Lane 41-- Conclusion

EVC2 mutation found in Patient 1 is not involved in abnormal BBSome complex.

2. Patients and Methods

Authors should disclose the approval Number from the Human Experimentation Committee.

3. Lanes 237-239

“Mice lacking in Bbs7 …as a result of abnormal IFT.”

Author should cite Reference #17 (Zhang et al.).

4. Lane 315

Fig. 5 (instead of Fig. 4) should be correct.

5. Figure 5

Authors should explain Fig. 5 properly in the text.

Figure 5 is too small to read. Figure should be enlarged.

There is no explanation about BBS2 mutation.

Reference should be written in refence numbers.

Author Response

Response to comments of Reviewer 2

Authors reported clinical case study of three patients containing EVC2, BBS2 and BBS7mutations, and predicted the mechanism of disease occurrence. The results seem to be interesting from the clinical point of view. However, the proposed mechanism is not conclusive from the viewpoint of experimental science. Authors should brush up the paper following the comments shown below. 

Major comments:

  1. Abstract

Lane 41-- Conclusion

EVC2 mutation found in Patient 1 is not involved in abnormal BBSome complex.

RESPONSE

Thank you for this comment. We corrected it as …. Conclusion: EVC2, BBS2, and BBS7 mutations found in our patients were implicated in malformation syndromes with dental anomalies including tooth agenesis, microdontia, taurodontism, and impaired dentin formation

  1. Patients and Methods

Authors should disclose the approval Number from the Human Experimentation Committee.

 RESPONSE

It is added as … The study was conducted in accordance with the Declaration of Helsinki and national guidelines. Informed consent was obtained from the parents in accordance with the regulations of the Human Experimentation Committee of the Faculty of Dentistry, Chiang Mai University (Certificate of Ethical Clearance number 25/2020).

  1. Lanes 237-239; “Mice lacking in Bbs7 …as a result of abnormal IFT.” Author should cite Reference #17 (Zhang et al.).

RESPONSE

Thank you for this comment. We added it as… Mice lacking in Bbs7 have primary cilia and IFT complexes [17], but there is an abnormal accumulation of dopamine D1 receptor to the ciliary membrane, as a result of abnormal IFT.

  1. Lane 315: Fig. 5 (instead of Fig. 4) should be correct.

  RESPONSE

We corrected it as…. Mutations in BBS disrupt retrograde IFT and disrupt transport of Inversin to the cytoplasm, preventing its interaction with DVL [27, 35] (Fig. 5). Thank you.

  1. Figure 5: Authors should explain Fig. 5 properly in the text.  Figure 5 is too small to read. Figure should be enlarged. Reference should be written in refence numbers.

RESPONSE

Thank you for this comment. I agree with you. We made figure 5 with the highest resolution possible. It is up to the journal. I hope the journal will have it enlarged as you suggested. References in figure 5 are now present in numbers. Thank you for the comments.

There is no explanation about BBS2 mutation.

RESPONSE

Thank you for this comment.

Round 2

Reviewer 1 Report

I would like to thank the authors for their efforts for improving the manuscript. It is excellent. I recommend this manuscript for publication without hesitation. I'm looking forward to reading the authors' future works. Best wishes!

Reviewer 2 Report

I admit the manuscript has been brushed up.

Authors should revise their Figure 5 as I advised in the previous comments.  Although they wrote "Flowchart shows EVC2, BBS2, and BBS7 mutations result in abnormal BBSome complex,...." as a title, we cannot find flow from BBS2 mutation in Figure 5. Authors modify Figure 5, and explain in the text